# The Hidden Link Between RLHF and Contrastive Learning

## Abstract

Alignment of large language models (LLMs) with human values has recently
garnered significant attention, with prominent examples including the canonical
yet costly Reinforcement Learning from Human Feedback (RLHF) and the sim-
ple Direct Preference Optimization (DPO). In this work, we demonstrate that
both RLHF and DPO can be interpreted from the perspective of mutual infor-
mation (MI) maximization, uncovering a profound connection to contrastive
learning. Within this framework, both RLHF and DPO can be interpreted as
methods that performing contrastive learning based on the positive and negative
samples derived from base model, leveraging the Donsker–Varadhan (DV) lower
bound on MI (equivalently, the MINE estimator). Such paradigm further reveals
why RLHF cannot correctly update when the probability of sampling the correct
answer from the base model is close to 0, which often leads to RLHF failing
to incentivize reasoning capacities in LLMs beyond what is already present in
the base model. Building on the perspective, we replace the DV/MINE bound
with the Jensen–Shannon (JS) MI estimator and propose the **M**utual **I**nformation
**O**ptimization (MIO). Comprehensive theoretical analysis and extensive empiri-
cal evaluations demonstrate that MIO mitigates the late-stage decline in chosen-
likelihood observed in DPO, achieving competitive or superior performance across
various challenging reasoning and mathematical benchmarks. The code is available
at: `https://anonymous.4open.science/r/MIO-63E6/`

## 1 Introduction

Recently, large language models (LLMs) have attained state-of-the-art performance on a wide
array of natural-language tasks (Bai et al., 2022). Aligning their outputs with human value is
commonly achieved through the methodology of reinforcement learning from human feedback
(RLHF) (Ouyang et al., 2022)(Stiennon et al., 2022), whose fine-tuning recipe has become one of the
industry standards. However, its dependence on two step reinforcement learning presents challenges
(Xiao et al., 2025), such as computational inefficiency and training instability. To mitigate these
limitations, alternative one-step approaches such as direct preference optimization (DPO) (Rafailov
et al., 2023) and its variants (Meng et al., 2024)(Tajwar et al., 2024) have been proposed , which has
successfully eliminated the need for explicit reward modeling. Specifically, an implicit reward based
on the likelihood of preference data is defined, which results in significant gains in efficiency while
preserving competitive performance.

> ┌─ **Major Questions** ─┐
> 1. *Could RLHF be theoretically framed as a form of contrastive learning?*
> 2. *Why do PPO/DPO fail to update when $\pi_\theta(y^*|x^*)$ is too small?*
> 3. *How can we overcome DPO's limitations to further improve the reasoning and mathematical performance of fine-tuned LLMs?*

Several recent works have hinted at conceptual links between RLHF and imitation learning (Ho &
Ermon, 2016). DIL (Xiao et al., 2025) show that optimizing a policy with RLHF implicitly maximizes
the likelihood of human-preferred outputs, formally proves that RLHF for aligning language models
could be view as an imitation learning problem. IRL(Wulfmeier et al., 2024) examines the pretraining
and fine-tuning of LLMs from an imitation learning standpoint, argue that standard next-token fine-
tuining is a form of imitation learning. Beyond imitation learning, several works have drawn intuitive

parallels between Direct Preference Optimization (DPO) and contrastive learning (Jaiswal et al., 2021). For example, Xu et al. (2024a) and Chen et al. (2024) treat DPO as a form of contrastive learning; However, these works lack theoretical depth and their analysis still lacks a formal theoretical foundation. Such observation motivates our central research question: **Is there a unified theoretical framework that rigorously establishes the connection between RLHF and contrastive learning?**

Recent empirical studies have challenged the widely held belief that *reinforcement learning with variable rewards* (RLVR) or standard KL-regularized RLHF can continuously unlock genuinely novel reasoning trajectories. Yue et al. (2025) observe that, on benchmarks such as GSM8K and MATH, reasoning chains produced after RLHF almost always fall within the sampling support of the *base* model. This suggests that RL primarily *amplifies* pre-existing capabilities rather than discovering new ones. However, the underlying causes of this phenomenon remain unclear. Additionally, some studies have pointed out that when the initial value of $\pi_\theta(y_l \mid x)$ is too small, DPO training tends to collapse. This implies that, whether for positive responses $y_w$ or negative responses $y_l$, if their probabilities become too small, RLHF ceases to be effective. This leads to our second question: **Why does RLHF (PPO/DPO) lose its ability to update when the probability of a specific answer $y^*$ sampled by the base model is too small?** This paper provides a rigorous theoretical explanation for these issues, which can ultimately be attributed to the Donsker–Varadhan mutual information estimator $I_{DV}$.

RLHF and its variants fundamentally adhere to a reward maximization objective, often instantiated through parametric models such as the Bradley–Terry (BT) model (Bradley & Terry, 1952). This modeling paradigm, while effective in aligning with preference data, has been shown to suffer from overfitting (Yuan et al., 2024; Pal et al., 2024), frequently leading to suboptimal generalization and misalignment with true user preferences (Xu et al., 2024b; Wu et al., 2024). Recent empirical studies further indicate that DPO and its variants progressively focus on unlearning the rejected responses. This process inadvertently increases the model's tendency to generate out-of-distribution responses, rather than reinforcing alignment with the chosen responses (Xu et al., 2024b). From a theoretical perspective, Yan et al. (2025) analyzes the optimization dynamics of DPO and demonstrates that the shared tokens between chosen and rejected responses lead to gradient interference. This interference introduces instability and a coupled degradation, whereby suppressing rejected responses inadvertently reduces the model's confidence in the selected responses. Such simultaneous unlearning of both rejected and chosen outputs ultimately degrades downstream performance—particularly in reasoning-intensive and mathematical tasks (Pal et al., 2024; Yuan et al., 2024; Meng et al., 2024). This observation motivates our third research question: **Can we design a principled algorithm that overcomes inherent shortcomings of DPO, while effectively enhancing model performance on complex reasoning and mathematical tasks?**

This paper makes the following contributions: (1) We present a unified mutual-information perspective on RLHF and contrastive learning, demonstrating that both RLHF and DPO optimize a Donsker–Varadhan (DV) (Donsker & Varadhan, 1983; Belghazi et al., 2018a) lower bound on mutual information, thereby aligning with contrastive objectives. (2) We provides a theoretical explanation for why RLHF cannot stimulate new reasoning paths in models, and why DPO fails to update effectively when $\pi_\theta(y \mid x)$ is too small: this is attributed to $I_{DV}$, which fails to provide the correct training signal to the policy model when $\pi_\theta(y \mid x)$ is too small. (3) Replaces the DV mutual information estimator with the JS mutual information estimator(Hjelm et al., 2019), resulting in a new method, MIO, which avoids the issue of synchronous descent and performs well on eight reasoning and mathematics-related problems. The paper rigorously proves that MIO does not encounter the problem of synchronous descent.

## 2 RETHINKING RLHF: A CONTRASTIVE LEARNING PERSPECTIVE

### 2.1 RLHF IS A FORM OF CONTRASTIVE LEARNING

**Answer to Question 1**: RLHF is a form of Contrastive Learning in Mutual Information frame.

Given a prompt $x = [x_1, x_2, ...]$, we are presented with **two candidate responses**: $y_w = [y_1, y_2, ...]$, the response preferred by humans; and $y_l$, the response with lower preference. Our goal is to fine-tuning the language model, enhancing its propensity to generate responses that are consistent with human preferences. Firstly, we assume that all positive examples $y_w$ are drawn from a preferred language model $\pi_{\text{chosen}}$, while negative examples $y_l$

are drawn from a dispreferred language model $\pi_{\text{rejection}}$. We assume that both $\pi_{\text{chosen}}$ and $\pi_{\text{rejection}}$ are energy-based models (EBMs) (Lecun et al., 2006), as discussed in Appendix A. Our objective is to adjust the target model $\pi_\theta$ to maximize mutual information with $\pi_{\text{chosen}}$ while minimizing mutual information with $\pi_{\text{rejection}}$.

We further reformulate this problem as a **contrastive learning objective**:

$$\max_\theta I(\pi_\theta, \pi_{\text{chosen}}) \qquad \text{and} \qquad \min_\theta I(\pi_\theta, \pi_{\text{rejection}}) \tag{1}$$

We begin by considering the unconstrained maximization of mutual information with respect to $\pi_{\text{chosen}}$, which reduces to a standard **imitation learning problem**:

$$\max_\theta I(\pi_{\text{chosen}}, \pi_\theta) \tag{2}$$

In Appendix B, we demonstrate that the contrastive learning and imitation learning objectives are equivalent, as improving one objective decreases the other and the strict local maximum of the former corresponding to the strict local minimum of the latter.

Since mutual information (MI) cannot be directly computed, we introduce an estimator for the lower bound of mutual information, namely the **M**utual **I**nformation **N**eural **E**stimation (MINE) Belghazi et al. (2018a):

$$I(\pi_\theta, \pi_{\text{chosen}}) = \sup_\phi \mathbb{E}_{P_{\pi_\theta \pi_{\text{chosen}}}}[T_\phi] - \log \mathbb{E}_{P_{\pi_\theta} P_{\pi_{\text{chosen}}}}[e^{T_\phi}], \tag{3}$$

where $P_{\pi_\theta \pi_{\text{chosen}}}$ denotes the joint distribution of $\pi_\theta$ and $\pi_{\text{chosen}}$, while $P_{\pi_\theta} P_{\pi_{\text{chosen}}}$ represents the marginal distribution of them. MINE maximizes the mutual information between $\pi_\theta$ and $\pi_{\text{chosen}}$ by optimizing a network-based function $T_\phi$.

Hence, we can derive the following expression for mutual information (see Appendix C.1):

$$I(\pi_\theta, \pi_{\text{chosen}}) = \sup_\phi \mathbb{E}_{P_{\pi_\theta \pi_{\text{chosen}}}}[T_\phi] - \log \left( \mathbb{E}_{P_{\pi_\theta}} \left[ \mathbb{E}_{P_{\pi_{\text{chosen}}}}[e^{T_\phi}] \right] \right) \tag{4}$$

To improve the optimization, we introduce a lower bound for the joint and marginal distributions:

$$I(\pi_\theta, \pi_{\text{chosen}}) \geq \sup_\phi \mathbb{E}_{P_{\pi_\theta \pi_{\text{chosen}}}}[T_\phi] - \log \left( \mathbb{E}_{P_{\pi_\theta}} \left[ 2\mathbb{E}_{P_{\bar{\pi}}}[e^{T_\phi}] \right] \right), \tag{5}$$

where $\bar{\pi}(\cdot|x) = \frac{1}{2}\pi_{\text{chosen}}(\cdot|x) + \frac{1}{2}\pi_{\text{rejection}}(\cdot|x)$. The right-hand side of equation equation 5 is also referred to as $I_{DV}$ (Hjelm et al., 2019). Considering the following Monte Carlo estimator:

$$\mathbb{E}_{\pi_\theta, \bar{\pi}} \left[ e^{T_\phi(\pi_\theta, \bar{\pi})} \right] \approx \frac{1}{2} \left( \frac{1}{M} \sum_{i=1}^M e^{T_\phi(\pi_\theta, \pi_{\text{chosen}})} + \frac{1}{N} \sum_{i=1}^N e^{T_\phi(\pi_\theta, \pi_{\text{rejection}})} \right). \tag{6}$$

We substitute equation 6 into equation 5, can estimate the lower bound of mutual information as:

$$\max_\phi \hat{I} = \log \left( \frac{e^{T_\phi(\pi_\theta, \pi_{\text{chosen}})}}{\frac{1}{M} \sum_{i=1}^M e^{T_\phi(\pi_\theta, \pi_{\text{chosen}})} + \frac{1}{N} \sum_{i=1}^N e^{T_\phi(\pi_\theta, \pi_{\text{rejection}})}} \right). \tag{7}$$

When $M = N = 1$, this objective could be simplified to:

$$\max_\phi \log \sigma \left( T_\phi(\pi_\theta, \pi_{\text{chosen}}) - T_\phi(\pi_\theta, \pi_{\text{rejection}}) \right), \tag{8}$$

where $T_\phi$ serves as a parameterized model used to estimate mutual information between two models, and is equivalent to the reward model in RLHF, see in Appendix A. Then, by treating $T_\phi$ as the reward model, we can derive an objective function that aligns precisely with the reward loss under the BT preference assumption in RLHF.

Furthermore, instead of retraining a separate neural network $T_\phi$, we constrain the T to the log-ratio sub-family (see in Appendix C.2):

$$T(\pi_\theta, \pi_{chosen}) = \log \frac{\pi_\theta(y_w|x)}{\pi_{\text{chosen}}(y_w|x)}, \qquad T(\pi_\theta, \pi_{rejection}) = \log \frac{\pi_\theta(y_l|x)}{\pi_{\text{rejection}}(y_l|x)}. \tag{9}$$

and substitute it into equation 8, we can arrive at the objective function of Direct Preference Optimization (DPO)(Rafailov et al., 2023):

$$\max_{\theta} \log \sigma \left( \beta \log \frac{\pi_\theta(y_w \mid x)}{\pi_{\text{ref}}(y_w \mid x)} - \beta \log \frac{\pi_\theta(y_l \mid x)}{\pi_{\text{ref}}(y_l \mid x)} \right) \qquad (10)$$

Keep constrain the T to the log-ratio sub-family equation 9. Only substituting this into the right-hand side of the MINE objective equation 4 and applying certain approximations yields the following objective used in the second stage of RLHF, where the policy model is updated using the reward model(Wu et al., 2024; Ghosh et al., 2020):

$$\max \mathbb{E}_{Joint}[T_\phi] - \text{KL}(\pi_\theta \parallel \pi_{\text{ref}}). \qquad (11)$$

The approximation error introduced here is upper-bounded (see Appendix E), and the bound depends on the ratio $\frac{\pi_\theta(\cdot|x)}{\pi_{\text{ref}}(\cdot|x)}$: the smaller this ratio, the smaller the approximation error. This offers us a new and insightful perspective about why PPO introduces a clipping function to constrain the update magnitude of the policy—large deviations in this ratio lead to large approximation errors, which in practice degrade performance. As the derivatives of $\hat{I}(\pi_\theta, \pi_{\text{chosen}})$ and $\hat{I}(\pi_\theta, \pi_{\text{rejection}})$ are opposites, maximizing $\hat{I}(\pi_{\text{chosen}}, \pi_\theta)$ simultaneously minimizes $\hat{I}(\pi_{\text{rejection}}, \pi_\theta)$. This shows that optimizing the mutual information between $\pi_\theta$ and $\pi_{\text{chosen}}$ inherently minimizes the mutual information between $\pi_\theta$ and $\pi_{\text{rejection}}$, thereby transforming the task into a contrastive learning framework.

Thus, we have derived the DPO objective function from a contrastive learning perspective by introducing a mutual information neural estimator (MINE), illustrating that DPO is essentially a special case of contrastive learning.

Furthermore, we can draw some insights:

1. Reward learning in RLHF is equivalent to the task of searching for a function $T \in \mathcal{F}_{\text{all}}$ within a specified function space. This function $T$ minimizes the discrepancy between estimated and actual unobservable mutual information (see Appendix C), thereby making the estimated and actual mutual information as close as possible.

2. Through constrain the T to the log-ratio sub-family $\mathcal{F}_{\text{sub}}$, after learning the reward model in RLHF, the subsequent update of the policy model is equivalent to a contrastive learning task. This phase can be seen as maximizing the mutual information between $\pi_\theta$ and the positive sample $\pi_{\text{chosen}}$, while minimizing the mutual information with the negative sample $\pi_{\text{rejection}}$. If, instead of training a separate parameterized neural network$T_\phi$, we restrict the search for $T$ to a subset $\mathcal{F}_{\text{sub}}$,we can derive the DPO loss function.

## 2.2 THE DV MUTUAL INFORMATION ESTIMATOR CAUSES RLHF COLLAPSE.

> **Answer to Question 2**: Why do PPO/DPO fail to update when $\pi_\theta(y^*|x^*)$ is too small? A mutual information explanation

In essence, based on our analysis, both RLHF and DPO can be viewed as approximate procedures for optimizing a DV/MINE-based mutual information estimator $I_{\text{DV}}(\theta)$. However, such estimators fail to provide meaningful gradient directions for a specific response $y^\star$ when the reference policy $\pi_{\text{ref}}(y^\star \mid x^\star)$ approaches zero. Since $\pi_\theta$ is typically initialized as a frozen copy of the reference model, if the initial $\pi_\theta(y^\star \mid x^\star)$ is close to zero, then no matter how well the reward model is trained, using it to further optimize $\pi_\theta$ becomes ineffective. **The root cause of this phenomenon is precisely the DV/MINE estimator** $I_{\text{DV}}(\theta)$. To formalize this claim, we present two theorems:

**Theorem 1 (DV/MINE Starvation Theorem).** If $\pi_{\text{ref}}(y^\star \mid x^\star) = 0$, then the DV/MINE mutual information estimator $I_{\text{DV}}(\theta)$ cannot guide any meaningful update for the specific response$y^\star$. Formally,

$$\langle \nabla_\theta I_{\text{DV}}(\theta), \nabla_\theta \log \pi_\theta(y^\star \mid x^\star) \rangle = 0.$$

This explains why PPO-style methods (e.g., GRPO) typically require a *cold start* stage prior to increase the probability of the base model sampling the correct answer $y^\star$: when the probability of sampling the correct answer $y^\star$ is too small, the cold start could increase it to prevent policy gradients from vanishing due to $\pi_{\text{ref}}(y^\star \mid x^\star) = 0$.

Some readers may object that, in practice, neural networks assign nonzero probability to every sequence (albeit extremely small), so Theorem 1 may not hold exactly. To address this, we provide a strengthened result:

**Theorem 2 (DV/MINE Starvation Theorem Pro).** As $\pi_\theta(y^\star \mid x^\star)$ approaches zero, the DV/MINE estimator $I_{\mathrm{DV}}(\theta)$ yields increasingly misleading gradients. Formally,

$$\left| \langle \nabla_\theta I_{\mathrm{DV}}(\theta), \, \nabla_\theta \log \pi_\theta(y^\star \mid x^\star) \rangle \right| \ \leq \ 2L \, \pi_\theta(y^\star \mid x^\star),$$

where $L$ is a constant. A full proof is provided in Appendix H.

This theorem offers a mutual-information perspective on why off-policy settings in PPO/GRPO often lead to degraded training performance: under an off-policy regime, the probability $\pi_\theta(y \mid x)$ of sampling a trajectory is extremely small, which makes the gradient of $I_{\mathrm{DV}}$ nearly orthogonal to the policy gradient. As a result, the mutual information estimator cannot be used to approximate the gradient of the policy model $\pi_\theta$. This also provides an intuitive insight: the effectiveness of DV/MINE-based gradient estimation is highly sensitive to the initial values of $\pi_\theta(y \mid x)$. Once these probabilities are too small, the estimator $I_{\mathrm{DV}}$ becomes *too hungry to update*.

In summary, the DV/MINE estimator introduces serious drawbacks: not only does it induce large gradient variance (see in Appendix D), but it also leads to *policy-gradient starvation* whenever $\pi_{\mathrm{ref}}(y \mid x)$ is near zero. This severely hinders the learning dynamics of RLHF. Mitigation strategies such as cold-start initialization are often used to increase $\pi_{\mathrm{ref}}(y \mid x)$, thereby reducing gradient starvation. However, the fundamental culprit remains the DV/MINE estimator itself. This motivates an important research question: **are there effective alternatives to DV/MINE-based mutual information estimators?**

## 3 METHODS: MUTUAL-INFORMATION OPTIMISATION

### 3.1 LET US REPLACE DV/MINE ESTIMATOR WITH JSD IN ALIGNMENT.

As our primary objective is to maximize Mutual Information rather than precisely estimate its value (Hjelm et al., 2019), we can rely on non-KL divergences, which offer favorable trade-offs. For instance, the Jensen-Shannon MI estimator (Nowozin et al., 2016)—similar to the binary cross-entropy used in minimizing total correlation (Brakel & Bengio, 2017)—is well-understood in neural network optimization and is more stable in practice compared to the DV-based objective (Hjelm et al., 2019):

$$I_{\mathrm{JSD}}(\pi_\theta; \pi_{\mathrm{chosen}}) := \sup_{\theta \in \Theta} \mathbb{E}_{\pi_\theta} \left[ -\mathrm{sp}(-T_\phi(\pi_\theta, \pi_{\mathrm{chosen}})) - \mathbb{E}_{\bar{\pi}}[\mathrm{sp}(T_\phi(\pi_\theta, \bar{\pi}))] \right]. \tag{12}$$

Considering the following Monte Carlo estimator:

$$\mathbb{E}_{\bar{\pi}} \left[ \mathrm{sp}(T_\phi(\pi_\theta, \bar{\pi})) \right] \approx \frac{1}{2} \Big( \frac{1}{M} \sum_{i=1}^{M} \log \left( 1 + \exp(T_\phi(\pi_\theta, \pi_{\mathrm{chosen}})) \right)$$

$$+ \frac{1}{N} \sum_{i=1}^{N} \log \left( 1 + \exp(T_\phi(\pi_\theta, \pi_{\mathrm{rejection}})) \right) \Big). \tag{13}$$

By replacing the DV/MINE (Belghazi et al., 2018a) surrogate $I_{\mathrm{DV}}$ in equation 4 with the JS-based surrogate $I_{\mathrm{JSD}}$ given in equation 12, and substituting equation 13 into equation 12, we obtain the following empirical objective:

$$\hat{I}_{\mathrm{JSD}} = -\frac{1}{M} \sum_{i=1}^{M} \log \left( 1 + e^{-T_\phi(\pi_\theta, \pi_{\mathrm{c}})} \right)$$

$$- \frac{1}{2} \left[ \frac{1}{M} \sum_{i=1}^{M} \log \left( 1 + e^{T_\phi(\pi_\theta, \pi_{\mathrm{c}})} \right) + \frac{1}{N} \sum_{j=1}^{N} \log \left( 1 + e^{T_\phi(\pi_\theta, \pi_{\mathrm{r}})} \right) \right]. \tag{14}$$

Setting $M = N = 1$ and constraining the critic to a log ratio family (See in Appendix C.2) gives a *closed-form* loss, which we call **Mutual-Information Optimisation (MIO)**:

$$\mathcal{L}_{\mathrm{MIO}}(\theta) \ = \ \log\!\Big(1 + \tfrac{\pi_{\mathrm{ref}}(y_w|x)}{\pi_\theta(y_w|x)}\Big) + \frac{1}{2}\log\!\Big(1 + \tfrac{\pi_\theta(y_w|x)}{\pi_{\mathrm{ref}}(y_w|x)}\Big) + \frac{1}{2}\log\!\Big(1 + \tfrac{\pi_\theta(y_l|x)}{\pi_{\mathrm{ref}}(y_l|x)}\Big). \tag{15}$$

## 3.2 Failure Mode of DPO when $\pi_\theta(y \mid x)$ is Close to Zero

Essentially, DPO estimates the policy gradient through the DV/MINE mutual information estimator. Consequently, it suffers from the same failure mode: when $\pi_\theta(y \mid x)$ becomes too small, the policy gradient vanishes. This effect is particularly pronounced for the rejected sample $y_l$ in preference alignment, since training explicitly encourages reducing its probability, making $\pi_\theta(y_l \mid x)$ more likely to approach zero over the course of optimization.

For a preference triple $(x, y_w, y_l)$ the DPO loss can be written as (Yan et al., 2025):

$$\ell_{\text{DPO}} = \log\Big(1 + \big[\alpha \, z^\beta\big]\Big), \qquad \alpha = \left[\frac{\pi_{ref}(y_w|x)}{\pi_{ref}(y_w|x)}\right]^\beta, \ z = \frac{\pi_\theta(y_l|x)}{\pi_\theta(y_l|x)}.$$

Denoting $\pi^+ = \pi_\theta(y_w \mid x)$ and $\pi^- = \pi_\theta(y_l \mid x)$, differentiation gives

$$\frac{\partial \ell_{\text{DPO}}}{\partial \pi^+} = -\frac{\alpha\beta}{1 + \alpha z^\beta} \frac{z^\beta}{\pi^+}, \qquad \frac{\partial \ell_{\text{DPO}}}{\partial \pi^-} = \frac{\alpha\beta}{1 + \alpha z^\beta} \frac{z^{\beta-1}}{\pi^+}.$$

Hence, we have that $|\partial_{\pi^-}\ell|/|\partial_{\pi^+}\ell| = \pi^+/\pi^-$. The gradient on the *rejected* probability therefore grows much faster than that on the *chosen* probability; as $\pi^- \to 0$ we have $\partial_{\pi^-}\ell \to \infty$ but $\partial_{\pi^+}\ell \to 0$. Because chosen and rejected completions usually share many tokens, the vanishing positive gradient cannot offset the dominant negative gradient, so the model is driven to lower *both* probabilities. This synchronous likelihood collapse—observed empirically in late-stage DPO training—degrades overall performance (Pal et al., 2024).

## 3.3 Stability of the MIO Gradient

**Answer to Question 3**: MIO leads to a more stable training process!

If the DV/MINE mutual information estimator causes gradient starvation when $\pi_\theta(y \mid x)$ is small, replacing it with a JSD-based mutual information estimator eliminates this issue. In other words, when $\pi_\theta(y \mid x)$ approaches zero, MIO does not suffer from vanishing policy gradients. This claim can be verified directly by differentiating the MIO loss. The MIO objective for the same triple is

$$\ell_{\text{MIO}} = \ln\big(1 + e^{-\beta\,\text{LR}^+}\big) + \tfrac{1}{2}\ln\big(1 + e^{\beta\,\text{LR}^+}\big) + \tfrac{1}{2}\ln\big(1 + e^{\beta\,\text{LR}^-}\big),$$

with $\text{LR}^\pm = \log \pi_\theta^\pm - \log \pi_{ref}^\pm$. Writing $\sigma^\pm = \sigma\big(\beta\,\text{LR}^\pm\big)$, one obtains

$$\partial_{\pi^+}\ell_{\text{MIO}} = \frac{\beta}{\pi^+}\big(1.5\sigma^+ - 1\big), \qquad \partial_{\pi^-}\ell_{\text{MIO}} = \frac{\beta}{2\pi^-}\sigma^-.$$

**Proposition 3.1** (Selective suppression of negatives). *Let* $\pi^- \to 0$*. Then, we have:*

$$\left|\frac{\partial \ell_{\text{MIO}}}{\partial \pi^-}\right| = \frac{\beta}{2\pi^-}\sigma^- \longrightarrow \infty, \qquad \left|\frac{\partial \ell_{\text{MIO}}}{\partial \pi^+}\right| = \frac{\beta}{\pi^+}\big|1.5\sigma^+ - 1\big|,$$

so the negative likelihood continues to be forcefully suppressed, while the positive gradient remains bounded and therefore *does not vanish*. Consequently, MIO preserves a strong corrective signal for rejected responses without impairing the optimisation of chosen responses.

**Proposition 3.2** (Self-regulating positive gradient). *Define* $\sigma^+ = \sigma\big(\beta\,\text{LR}^+\big)$*. Then, we have:*

$$\partial_{\pi^+}\ell_{\text{MIO}} = \frac{\beta}{\pi^+}\big(1.5\sigma^+ - 1\big) \begin{cases} > 0 & \text{if } \sigma^+ > \frac{2}{3}, \\ < 0 & \text{if } \sigma^+ < \frac{2}{3}. \end{cases}$$

Hence, MIO exhibits an *intrinsic clamping mechanism*: once a chosen token attains excessive probability ($\sigma^+ > 2/3$) the gradient turns positive, gently pushing its likelihood downward and mitigating over-fitting; conversely, if the token remains under-represented ($\sigma^+ < 2/3$) the gradient is negative, encouraging further amplification. This self-adjusting, bi-phase behaviour amounts to an implicit *self-paced curriculum*: optimisation effort is automatically reallocated from already-well-learned positives to the more informative, harder ones, yielding superior robustness and sample efficiency compared with DPO.

# 4 EXPERIMENT

## 4.1 DPO BREAKS WHEN $\pi_\theta(y_l \mid x)$ NEAR 0, BUT MIO NOT!

The previous section theoretically demonstrated that methods dependent on $I_{DV}(\theta)$, such as DPO and PPO, encounter issues with policy gradients becoming ineffective when $\pi_\theta(y \mid x)$ approaches zero, especially for $\pi_\theta(y_l \mid x)$. This is particularly evident because $\pi_\theta(y_l \mid x)$ is more likely to approach zero during training than $\pi_\theta(y_w \mid x)$. However, this problem does not occur in the MIO approach. The earlier discussion was purely theoretical, and while it is challenging to validate this in practice, we have prepared a toy model experiment to empirically investigate this issue.

Following (Yan et al., 2025), the toy model setup, shown in Figure 1, consists of a discrete space with 4 prompts and 10 responses. The policy $\pi_\theta$, implemented as a three-layer MLP, processes a one-hot input vector and outputs a categorical distribution over the responses, which are divided into three categories: selected responses (first 4 dimensions), rejected responses (next 4 dimensions), and unseen responses (final 2 dimensions). Each prompt has a corresponding optimal response (e.g., response 1 is optimal for prompt 1). In DPO or MIO training, preference data is generated through a mini-batch sampling strategy, where an ideal annotator matches each prompt with its optimal response, creating a diagonal preference matrix. In each mini-batch, one additional response is randomly selected to form preference pairs, ensuring diverse gradient updates.

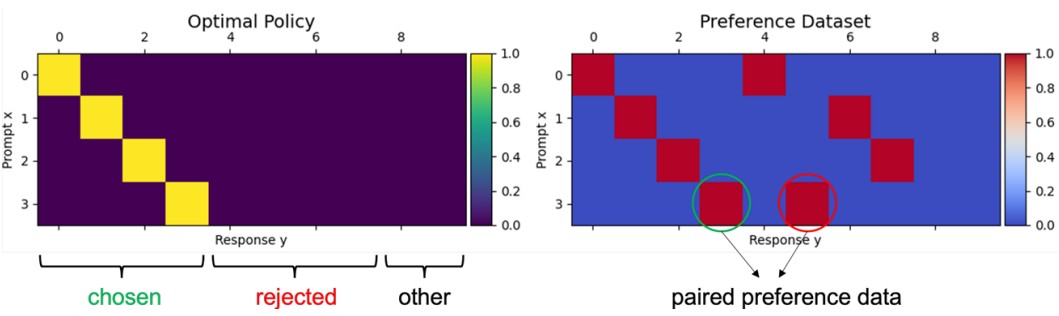

Figure 1: Toy model setup(Yan et al., 2025). Left: the optimal policy where the highlighted blocks represent optimal responses. Right: preference dataset construction.

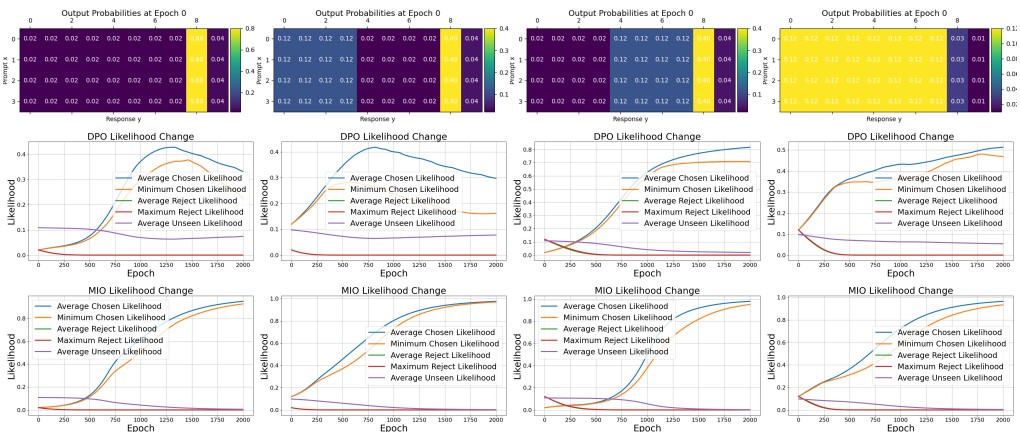

Figure 2: Overview of the DPO and MIO dynamics.From left to right, the figures show the initial state and the likelihood dynamics for chosen/rejected/unseen responses in Scenarios 1 to 4, similar to the left diagram in Figure 2: (1) both chosen and rejected responses are very small, (2) chosen is normal and rejected is very small, (3) chosen is small and rejected is normal, and (4) both chosen and rejected is normal.

Figure 2 contrasts the learning dynamics of DPO with those of our proposed MIO objective. When the probabilities of both the *chosen* and *rejected* responses remain in a moderate range, DPO converges smoothly and the likelihood of the chosen response declines only marginally relative

to that of the rejected one. Once the rejected–response probability becomes very small, however, DPO develops a pathological *synchronous collapse*: the probabilities of *both* the chosen and the rejected responses decrease in lock-step. Across all regimes we tested, MIO displays no such joint degradation. When the rejected probability is pushed towards zero, DPO and its derivatives exhibit synchronous decline of the chosen and rejected likelihoods, mirroring what is observed in practice (Yan et al., 2025; Chen et al., 2024; Ho & Ermon, 2016), **whereas MIO remains entirely immune to this failure mode**. A formal explanation of this qualitative difference is provided in Section 3.2.

The toy model serves as an abstract simulation that amplifies the 3D-properties of DPO (Yan et al., 2025), which leads to a decrease in the likelihood of both chosen and rejected responses. However, MIO does not exhibit this effect. While the toy model setup differs significantly from real-world LLM training—such as in sampling frequency—it provides valuable insights. In real-world scenarios, DPO and MIO are typically trained over one epoch, with each data point used only a few times. In contrast, the toy model samples the same data points repeatedly. Conceptually, this is akin to treating each input/output as a token rather than a full prompt/response, where each token may be sampled multiple times during real-world training.

## 4.2 REAL WORLD EXPERIMENTAL SETUP

We fine-tune both Mistral-7B-SFT(Tunstall et al., 2023a) and LLaMA3-8B-SFT on the 64k-prompt *UltraFeedback-Binarized* (Cui et al., 2024) corpus, as well as train directly on Qwen2.5-7B-Instruct, where each prompt is paired with a *chosen* and a *rejected* completion selected by GPT-4, yielding 64k positive–negative pairs; we retain the official train/validation split of *UltraFeedback-Binarized*.

Model quality is assessed on eight public mathematical and reasoning suites: Hendrycks MATH (Hendrycks et al., 2021a) (12.5 k competition-level problems plus the level-5 "hard" subset), Minerva Math (Lewkowycz et al., 2022a) (quantitative-reasoning items distilled from GSM8K and MATH), MultiMedQA (Singhal et al., 2022) (six medical QA benchmarks spanning USMLE, PubMedQA, and consumer health queries), MathQA (Amini et al., 2019a) (37 k multiple-choice word problems with executable rationales), GSM8K (Cobbe et al., 2021a) (8.5 k grade-school arithmetic questions), AQuA-RAT (Ling et al., 2017) (100 k algebraic problems with free-text solutions), and MuSR (Sprague et al., 2024) (narrative tasks requiring multistep soft reasoning over 700–1000-word stories). The code, hyperparameter settings, and evaluation metrics is provid in the Appendix G.

The result is in table 1. For Mistral-7B-SFT, MIO achieves the best performance on five tasks and ranks second on two others across the eight mathematical and reasoning benchmarks. For LLaMA3-8B-SFT, MIO ranks first in seven of the eight benchmarks and second in one. For Qwen2.5-7B-Instruct, MIO achieves the best performance across all eight benchmarks. These results demonstrate that MIO offers superior generalization in reasoning and mathematical problem-solving compared to existing alignment methods.

## 4.3 MIO PREVENTS THE CHOSEN-REWARD COLLAPSE

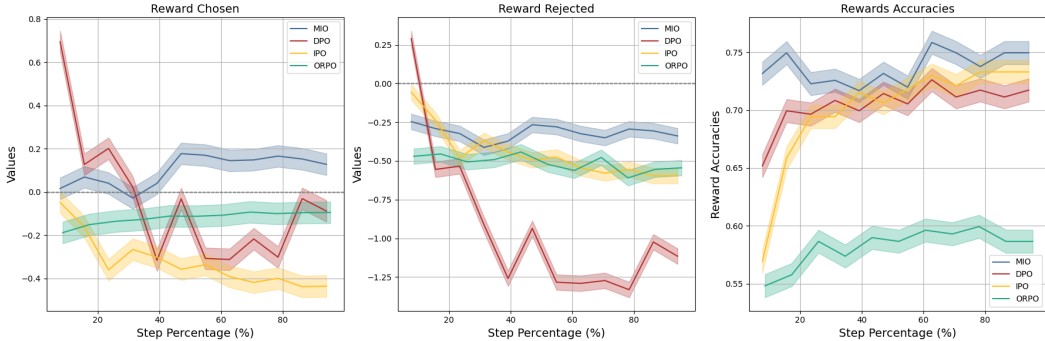

Figure 3: Training curves on MISTRAL-7B-SFT. Unlike DPO, IPO, and ORPO, MIO *increases* the rewards of chosen responses while suppressing rejected ones, thereby avoiding the "synchronous collapse" observed in prior methods.

Table 1: Evaluation results on 8 benchmarks. The **best** and second best are marked.

| Model (↓) / Benchmark (→) | | Hendrycks Math | Minerva Math | Multimedqa | MathQA | GSM8K | Aqua Rat | Math Hard | MUSR |
|---|---|---|---|---|---|---|---|---|---|
| | SFT | 0.14 | 8.80 | 49.03 | 29.91 | 32.29 | 19.29 | 2.72 | 39.94 |
| Mistral-7B-SFT | DIL(Xiao et al., 2025) | 0.04 | 6.46 | 49.54 | 28.94 | 28.35 | 21.33 | 2.11 | 38.36 |
| | DPO(Rafailov et al., 2023) | 0.04 | 8.14 | 49.21 | 30.72 | 36.47 | 19.69 | 2.72 | **43.65** |
| | IPO(Azar et al., 2023) | 0.08 | 6.50 | 48.39 | 29.41 | 3.71 | 19.69 | 0.23 | 35.71 |
| | KTO(Ethayarajh et al., 2024) | 0.04 | **10.38** | 49.76 | 30.56 | **41.75** | 21.65 | 3.43 | 43.39 |
| | NCA(Chen et al., 2024) | 0.14 | 8.92 | 49.58 | 30.59 | 37.68 | 18.50 | 2.49 | 39.68 |
| | ORPO(Hong et al., 2024) | 0.04 | 9.68 | 49.40 | 29.78 | 36.39 | 24.80 | 2.57 | 42.06 |
| | SimPO(Meng et al., 2024) | 0.19 | 8.36 | 49.16 | 30.93 | 39.20 | 18.11 | 2.49 | 39.02 |
| | SLiC(Zhao et al., 2023) | 0.08 | 8.72 | 49.74 | 30.66 | 39.73 | 20.47 | 3.17 | 42.99 |
| | CPOXu et al. (2024a) | 0.16 | 7.14 | 49.18 | 27.57 | 31.59 | 23.37 | 3.40 | 42.44 |
| | MIO | **0.21** | 9.08 | **49.82** | **30.96** | 40.16 | **24.83** | **3.57** | 43.42 |
| | SFT | 0.06 | 14.06 | 56.84 | 28.07 | 50.26 | 25.90 | 4.61 | 39.94 |
| LLaMA3-8B-SFT | DIL(Xiao et al., 2025) | 0.19 | 17.16 | 57.01 | 34.85 | 55.86 | 23.62 | 5.06 | 40.75 |
| | DPO(Rafailov et al., 2023) | 0.04 | 16.48 | 55.54 | 28.17 | 54.38 | 23.62 | 4.23 | 39.11 |
| | IPO(Azar et al., 2023) | 0.12 | 15.26 | 56.99 | 29.85 | 53.07 | 25.69 | 4.91 | 36.77 |
| | KTO(Ethayarajh et al., 2024) | 0.04 | 16.42 | 31.36 | 19.83 | 55.57 | 22.05 | 5.03 | 35.05 |
| | NCA(Chen et al., 2024) | 0.08 | 17.24 | **57.53** | 28.41 | 54.81 | 25.20 | 4.46 | 38.62 |
| | ORPO(Hong et al., 2024) | 0.08 | 16.86 | 53.81 | 28.01 | 48.98 | 25.59 | 4.68 | 38.76 |
| | SimPO(Meng et al., 2024) | 0.06 | 17.16 | 57.13 | 36.35 | 53.37 | 24.80 | 4.61 | 41.08 |
| | SLiC(Zhao et al., 2023) | 0.17 | 15.18 | 51.94 | 28.58 | 46.22 | 26.77 | 3.59 | 38.03 |
| | CPOXu et al. (2024a) | 0.13 | 14.26 | 54.01 | 26.13 | 52.16 | 25.39 | 3.85 | 39.15 |
| | MIO | **0.20** | **17.68** | 57.32 | **36.85** | **56.86** | **27.62** | **5.44** | **42.81** |
| | Instruct | 0.04 | 25.20 | 55.54 | 33.33 | 71.19 | 21.65 | 49.47 | 38.5 |
| Qwen-7B-Instruct | DIL(Xiao et al., 2025) | 0.12 | 23.12 | 55.31 | 33.66 | 72.21 | 24.80 | 46.22 | 37.29 |
| | DPO(Rafailov et al., 2023) | 0.11 | 26.16 | 57.36 | 33.21 | 73.46 | 22.83 | 50.08 | 39.55 |
| | IPO(Azar et al., 2023) | 0.18 | 25.44 | 56.28 | 34.20 | 71.57 | 21.26 | 50.38 | 40.08 |
| | KTO(Ethayarajh et al., 2024) | 0.19 | 26.91 | 52.86 | 29.7 | 75.08 | 20.08 | 46.00 | 37.99 |
| | NCA(Chen et al., 2024) | 0.07 | 25.34 | 56.61 | 34.30 | 72.56 | 22.05 | 49.47 | 39.81 |
| | ORPO(Hong et al., 2024) | 0.02 | 26.34 | 55.49 | 37.98 | 74.30 | 28.58 | 32.18 | 39.97 |
| | SimPO(Meng et al., 2024) | 0.19 | 26.6 | 45.25 | 34.80 | 71.72 | 25.98 | 38.44 | 37.96 |
| | SLiC(Zhao et al., 2023) | 0.06 | 27.54 | 57.32 | 36.31 | 73.99 | 25.98 | 46.07 | 38.76 |
| | CPOXu et al. (2024a) | 0.02 | 18.76 | 54.76 | 36.14 | 72.71 | 28.74 | 46.27 | 37.17 |
| | MIO | **0.22** | **27.84** | **57.61** | **38.17** | **75.39** | **30.16** | **51.36** | **40.34** |

During DPO training, the rewards of *all* responses ($y_w$ and $y_l$) often decrease simultaneously, a harmful "unlearning" effect that degrades downstream performance (Xiao et al., 2025; Meng et al., 2024; Pal et al., 2024). Figure 3 shows that the proposed MIO objective completely eliminates this failure mode: the chosen-rewards increase while the rejected rewards drop slightly. Section 3.2 provides a theoretical explanation of this behavior.

Some readers may argue that NCA-Pair can also increase the chosen reward while decreasing the rejected reward. To examine this, Figure 5 (in Appendix) presents a direct comparison between MIO and NCA-Pair in terms of reward margin and related metrics. The results show that MIO achieves a greater increase in both the chosen reward and the reward margin, as well as a larger decrease in the rejected reward, compared to NCA-Pair. This indicates that MIO leverages the reward signal more effectively than NCA-Pair.

## 5 CONCLUSION

In this work, we propose a novel perspective that frames RLHF and DPO as a unified form of contrastive learning over the distributions of chosen and rejected responses, grounded in mutual information. Building on this connection, we provide rigorous explanations for several phenomena observed in prior work. For example, reasoning paths that are not sampled by the base model remain inaccessible even after RL training, and DPO fails to update effectively when the probability of a rejected response becomes too small. We show that all of these issues can be attributed to the DV/MINE mutual information estimator. Furthermore, we introduce the **M**utual **I**nformation **O**ptimization (MIO), replacing the Donsker-Varadhan (DV) bound with a Jensen-Shannon (JSD) bound. MIO demonstrates superior preservation of reasoning and mathematical abilities and offers greater training stability compared to existing baselines. Extensive experimental results demonstrate that MIO achieves superior performance on a wide range of mathematical and reasoning benchmarks. We hope that our work could offer useful insights for the community and help address the limitations of current RLHF methods. This work does not explore the effects of alternative mutual information estimators, which we leave as a promising direction for future research. Details regarding LLM usage are provided in the appendix J.

## ETHICS STATEMENT

This work adheres to the ICLR Code of Ethics. No human-subject or animal experiments were conducted. All datasets used, including **UltraFeedbackBinarized** (Cui et al., 2023a; Tunstall et al., 2023b), were obtained and used in compliance with their licenses and applicable policies. We took precautions to avoid privacy risks and discriminatory outcomes. No personally identifiable information was processed, and no experiments posed security or safety concerns. We are committed to transparency and integrity throughout the research lifecycle.

## REPRODUCIBILITY STATEMENT

We took extensive steps to ensure the results are reproducible. We release code, configuration files, and processing scripts in an anonymized repository to support replication. The experimental setup—including training procedures, model architectures, hyperparameters, and hardware details—is documented in the paper and the repository README. We also provide a detailed description of experiment details G to facilitate faithful reimplementation. Furthermore, we have released all code associated with training and experiments, which is publicly available at `https://anonymous.4open.science/r/MIO-63E6/` for full reproducibility.

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

## A  SEARCHING FOR A CRITIC $T_\phi$ IS EQUIVALENT TO TRAINING A REWARD MODEL.

**Energy–based form of the *chosen* model.**  Throughout the analysis we assume that the preference-filtered policy $\pi_{\text{chosen}}$ is obtained from a frozen *reference* model $\pi_{\text{ref}}$ via an energy re-weighting:

$$\pi_{\text{chosen}}(y \mid x) \;=\; \pi_{\text{ref}}(y \mid x)\, \frac{e^{\alpha\, r(x,y)}}{Z(x)}, \tag{A.1}$$

where $r(x,y)$ is a reward signal, $\alpha > 0$ a temperature, and $Z(x) = \mathbb{E}_{y \sim \pi_{\text{ref}}} e^{\alpha r(x,y)}$ the normaliser.

**Definition A.1.** *For a positive pair $(x, y_w)$ we define*

$$T_\phi(\pi_\theta, \pi_{chosen}) = \log \frac{\pi_\theta(y_w \mid x)}{\pi_{chosen}(y_w \mid x)}, \tag{A.2}$$

$$r(x, y_w) = \beta\Big[\log \tfrac{\pi_\theta(y_w|x)}{\pi_{ref}(y_w|x)} + \log Z(x)\Big], \tag{A.3}$$

*with a scaling constant $\beta > 0$.*

**Substitute equation A.3 into equation A.1.**

$$\pi_{\text{chosen}}(y_w \mid x) = \pi_{\text{ref}}^{1-\alpha\beta}\, \pi_\theta^{\alpha\beta}\, Z(x)^{\alpha\beta-1}. \tag{A.4}$$

**Insert equation A.4 into equation A.2.**

$$T_\phi(\pi_\theta, \pi_{\text{chosen}}) = (1 - \alpha\beta)\Big[\log \pi_\theta(y_w \mid x) - \log \pi_{\text{ref}}(y_w \mid x) + \log Z(x)\Big]. \tag{A.5}$$

Which is to say:

$$T_\phi(\pi_\theta, \pi_{\text{chosen}}) = \log \frac{\pi_\theta(y_w \mid x)}{\pi_{\text{chosen}}(y_w \mid x)} \propto \log \frac{\pi_\theta(y_w \mid x)}{\pi_{\text{ref}}(y_w \mid x)} \tag{A.6}$$

**Compact form.**  Introducing a single scaling parameter

$$\gamma \;:=\; \frac{1 - \alpha\beta}{\beta}, \tag{A.7}$$

and using equation A.3 in equation A.5 we arrive at

$$\boxed{T_\phi(\pi_\theta, \pi_{\text{chosen}}) \;=\; \gamma\, r(x, y_w)} \tag{A.8}$$

for the positive pair, and analogously

$$\boxed{T_\phi(\pi_\theta, \pi_{\text{rejection}}) \;=\; \gamma\, r(x, y_l)} \tag{A.9}$$

for the negative pair $(x, y_l)$.

## B  MAXIMIZING MUTUAL INFORMATION TOWARDS THE POSITIVE MODEL INHERENTLY DISTANCES IT FROM THE NEGATIVE MODEL.

**Definition B.1.** *For every prompt $x$ draw $y_w \sim \pi_{chosen}(\cdot \mid x)$ (positive) and $y_l \sim \pi_{rejection}(\cdot \mid x)$ (negative). Fix a critic $T_\phi \in \mathcal{F}$. With $z^+ = T_\phi(x, y_w)$ and $z^- = T_\phi(x, y_l)$, define the two **DV / InfoNCE** estimators:*

$$\hat{I}^+ = \log \frac{e^{z^+}}{e^{z^+} + e^{z^-}} = -\log\big(1 + e^{-(z^+ - z^-)}\big), \tag{B.1}$$

$$\hat{I}^- = \log \frac{e^{z^-}}{e^{z^+} + e^{z^-}} = -\log\big(1 + e^{\,z^+ - z^-}\big). \tag{B.2}$$

Write the **logit gap** $\Delta := z^+ - z^-$. With $\sigma(u) = \frac{1}{1+e^{-u}}$ we have $\hat{I}^+ = \log \sigma(\Delta)$, $\hat{I}^- = \log \sigma(-\Delta)$.

**Gradients share the $\Delta$ direction but opposite sign.**

$$\frac{\partial \hat{I}^+}{\partial \Delta} = \sigma(-\Delta), \qquad \frac{\partial \hat{I}^-}{\partial \Delta} = -\sigma(\Delta). \tag{B.3}$$

Because $\sigma(u) > 0 \;\forall u \in \mathbb{R}$,

$$\operatorname{sign}\!\left(\partial_\Delta \hat{I}^+\right) = -\operatorname{sign}\!\left(\partial_\Delta \hat{I}^-\right). \tag{B.4}$$

**Coupled parameter gradients.** Let the model parameters be $\theta$ and assume $z^+, z^-$ are differentiable in $\theta$. By the chain rule

$$\nabla_\theta \hat{I}^+ = \frac{\partial \hat{I}^+}{\partial \Delta}\,\nabla_\theta \Delta, \qquad \nabla_\theta \hat{I}^- = \frac{\partial \hat{I}^-}{\partial \Delta}\,\nabla_\theta \Delta. \tag{B.5}$$

Combine equation B.3–equation B.5:

$$\nabla_\theta \hat{I}^- = -\,\frac{\sigma(\Delta)}{\sigma(-\Delta)}\,\nabla_\theta \hat{I}^+. \tag{B.6}$$

The scalar factor $-\sigma(\Delta)/\sigma(-\Delta) \,<0$; hence

$$\boxed{\left\langle \nabla_\theta \hat{I}^+,\, \nabla_\theta \hat{I}^- \right\rangle < 0} \tag{B.7}$$

i.e. the two gradient vectors always point in **opposite directions**.

**Consequence (first-order).** Any update $\theta \leftarrow \theta + \eta \nabla_\theta \hat{I}^+$ ($\eta > 0$) that **increases** the positive MI estimator must **decrease** the negative MI estimator:

$$\hat{I}^+(\theta_{\text{new}}) > \hat{I}^+(\theta) \implies \hat{I}^-(\theta_{\text{new}}) < \hat{I}^-(\theta). \tag{B.8}$$

SECOND-ORDER COUPLING: EXTREMA COINCIDE

**Shared critical points.** From equation B.6, $\nabla_\theta \hat{I}^+ = 0$ iff $\nabla_\theta \hat{I}^- = 0$; thus both objectives share the same set of stationary points.

**Hessian analysis.** Let $f(\theta) = \hat{I}^+(\theta) = h(\Delta(\theta))$ and $g(\theta) = \hat{I}^-(\theta) = h(-\Delta(\theta))$ with $h(u) = \log \sigma(u)$. Then

$$h'(u) = \sigma(-u) > 0, \qquad h''(u) = -\sigma(u)\sigma(-u) < 0.$$

Using standard composition rules,

$$\nabla_\theta^2 f = h''(\Delta)\,\nabla\Delta\nabla\Delta^\top + h'(\Delta)\,\nabla^2\Delta, \tag{B.9}$$

$$\nabla_\theta^2 g = h''(-\Delta)\,\nabla\Delta\nabla\Delta^\top - h'(-\Delta)\,\nabla^2\Delta. \tag{B.10}$$

At any common critical point $\theta^\star$ we have $\nabla\Delta(\theta^\star) = 0$; the rank-one terms vanish, leaving

$$\nabla_\theta^2 f(\theta^\star) = h'(\Delta^\star)\,\nabla^2\Delta(\theta^\star), \qquad \nabla_\theta^2 g(\theta^\star) = -h'(-\Delta^\star)\,\nabla^2\Delta(\theta^\star). \tag{B.11}$$

Because $h'(\cdot) > 0$, the two Hessians differ *only* by an overall sign.

**Extremum equivalence.** If $\nabla_\theta^2 f(\theta^\star) \prec 0$ (negative definite), then $\nabla_\theta^2 g(\theta^\star) \succ 0$ (positive definite); hence

$$\boxed{\begin{aligned} &\theta^\star \text{ is a } \textit{strict} \text{ local maximum of } \hat{I}^+ \\ &\iff \theta^\star \text{ is a } \textit{strict} \text{ local minimum of } \hat{I}^-. \end{aligned}} \tag{B.12}$$

**Implication for optimisation.** Therefore,

$$\arg\max_\theta \hat{I}^+(\theta) \;=\; \arg\min_\theta \hat{I}^-(\theta),$$

provided the optimum is strict and isolated. Gradient-based methods that converge to a strict maximiser of $\hat{I}^+$ will, by construction, converge to the corresponding strict minimiser of $\hat{I}^-$, rigorously validating the coupling intuition expressed in the main text.

# C  MUTUAL-INFORMATION ESTIMATION: PRACTICAL SURROGATE AND CONSTRAIN CRITIC T

## C.1  DV / MINE SAMPLING SCHEME FOR RLHF

**Modified DV bound.**   Under these measures the Donsker–Varadhan (DV) / MINE lower bound reads

$$I(\pi_\theta, \pi_{\text{chosen}}) = \sup_\phi \mathbb{E}_{P_{\pi_\theta \pi_{\text{chosen}}}}[T_\phi] - \log(\mathbb{E}_{P_{\pi_\theta}} \mathbb{E}_{P_{\pi_{\text{chosen}}}}[e^{T_\phi}]) \tag{C.1}$$

$$\geq \sup_\phi \mathbb{E}_{P_{\pi_\theta \pi_{\text{chosen}}}}[T_\phi] - \log(\mathbb{E}_{P_{\pi_\theta}} \{\mathbb{E}_{P_{\pi_{\text{chosen}}}}[e^{T_\phi}] + \mathbb{E}_{P_{\pi_{\text{rejection}}}}[e^{T_\phi}]\}) \tag{C.2}$$

$$= \sup_\phi \mathbb{E}_{P_{\pi_\theta \pi_{\text{chosen}}}}[T_\phi] - \log(\mathbb{E}_{P_{\pi_\theta}} \{2\mathbb{E}_{P_{\bar\pi}}[e^{T_\phi}]\}) \tag{C.3}$$

$$= \sup_\phi \mathbb{E}_{P_{\pi_\theta \pi_{\text{chosen}}}}[T_\phi] - \log(\mathbb{E}_{P_{\pi_\theta} P_{\bar\pi}}[e^{T_\phi}]) - \log 2 \tag{C.4}$$

$$= \sup_\phi \mathbb{E}_{P_{\pi_\theta \pi_{\text{chosen}}}} \log e^{[T_\phi]} - \log(\mathbb{E}_{P_{\pi_\theta} P_{\bar\pi}}[e^{T_\phi}]) - \log 2 \tag{C.5}$$

## C.2  LET'S CONSTRAIN T TO A RESTRICTED CRITIC FAMILY

If we therefore constrain the critic to the log-ratio sub-family

$$\mathcal{F}_{\text{sub}} = \left\{ T(\pi_\theta, \pi_c) = \log \frac{\pi_\theta(y|x)}{\pi_c(y|x)} + c \ \middle| \ c \in \mathbb{R} \right\}, \tag{C.6}$$

whose evaluation requires only one forward pass through $\pi_c$.

Plugging log-ratio $T$ into equation C.5 yields the *surrogate* bound

$$\hat{I}(\theta) := \sup_\theta \left[ \mathbb{E}_{P_{\pi_\theta}, \pi_c}[T(\pi_\theta, \pi_c)] - \log \mathbb{E}_{P_{\pi_\theta} P_{\bar\pi}}[e^{T(\pi_\theta, \bar\pi)}] \right] - \log 2, \tag{C.7}$$

which satisfies

$$\boxed{\hat{I}(\theta) \ \leq \ I_{\bar\pi}^\star(\theta) \ \leq \ I(\pi_\theta, \pi_c)}. \tag{C.8}$$

The first inequality reflects the restriction $\mathcal{F}_{\text{sub}} \subset \mathcal{F}_{\text{all}}$; the second arises from replacing $\pi_c$ by $\bar\pi$ in the denominator. Despite this looseness, Appendix C shows that the positive and negative gradients are always anti-parallel, guaranteeing a contrastive signal at every optimisation step.

# D  THE JSD ESTIMATOR YIELDS SUBSTANTIALLY LOWER VARIANCE THAN DV/MINE

Subsequent research has shown that the DV-based objective used in MINE suffers from significant instability in practice (Poole et al., 2019). Specifically, its variance can grow exponentially with the true mutual information (MI) (Song & Ermon, 2020), leading to exploding gradients and requiring very large batch sizes to maintain stable training (Guo et al., 2022). Poole et al. (2019) further demonstrate that DV-style bounds like MINE exhibit low bias but extremely high variance, causing erratic gradient updates.

In the work (McAllester & Stratos, 2020), it is emphasized that accurate MI estimation is intrinsically challenging: high bias or high variance can lead to overfitting of spurious correlations. Thus, in contrastive learning, stable MI lower bounds (e.g., MINE or the Jensen–Shannon (JS) bound) should be prioritized over maximizing numerical MI estimates.

Additionally, Sordoni et al. (2021) point out that the MINE bound is upper-bounded by $\log K$, where $K$ is the number of negative samples. When the true MI greatly exceeds $\log K$, the bound severely underestimates MI.

Collectively, these studies (Guo et al., 2022; McAllester & Stratos, 2020) highlight that in regimes with few negative samples, MINE suffers from prohibitively large variance (Poole et al., 2019),

leading to unstable gradients. Therefore, replacing MINE with a variance-reduced alternative, such as the JSD estimator (Wen et al., 2020), is crucial for reliable optimization.

We investigate the behavior of mutual information (MI) estimators in the low-negative-sample regime using a synthetic bivariate Gaussian setting, where the correlation coefficient $\rho$ controls the dependency between variables. The ground-truth MI is given analytically as $I(X, Y) = -\frac{1}{2} \log(1 - \rho^2)$.

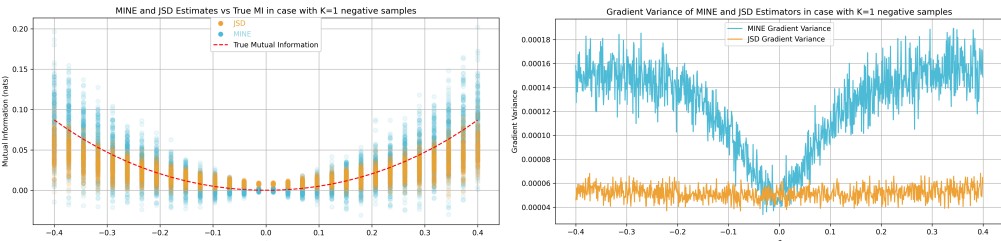

Figure 4: Left: Estimated mutual information under the bivariate Gaussian setting with varying $\rho$, using MINE and JSD estimators with $K = 1$ negative sample. While both methods track the ground-truth trend, JSD is consistently than MINE. Right: Gradient variance of each estimator. The JSD estimator yields substantially lower variance, indicating more stable optimization behavior under limited negative sampling.

The left panel of Figure 4 compares the DV/MINE and JSD estimators with a single negative sample ($K = 1$). While both estimators qualitatively capture the ground-truth MI trend, JSD does not consistently outperform MINE in terms of accuracy. However, **when gradient stability is prioritized over precise MI estimation, JSD demonstrates a clear advantage** (Shrivastava et al., 2023). To further investigate this, we analyze the gradient variance induced by each estimator, with the right panel of Figure 4 showing the gradient variance computed from the first-layer weights of a three-layer MLP trained for each estimator's objective. Under conditions with limited negative samples, **JSD exhibits significantly greater stability than MINE**, making it particularly suitable for training objectives, such as Alignment, that rely on contrastive MI estimates with sparse negative sampling (Chen et al., 2024).

# E    UPPER BOUNDS ON THE JENSEN GAP

We consider the approximation

$$\mathbb{E}_{\mathrm{J}}\big[T_\phi\big] \ - \ \log \mathbb{E}_{\pi_\theta}\Big[\frac{\pi_\theta}{\pi_{\mathrm{chosen}}}\Big] \ \approx \ \mathbb{E}_{\mathrm{J}}\big[T_\phi\big] \ - \ \mathbb{E}_{\pi_\theta}\Big[\log \frac{\pi_\theta}{\pi_{\mathrm{chosen}}}\Big],$$

and denote the resulting error (Jensen gap) by

$$\Delta \ = \ \mathbb{E}_{\pi_\theta}\big[\log f(Y)\big] \ - \ \log \mathbb{E}_{\pi_\theta}\big[f(Y)\big], \qquad f(Y) = \frac{\pi_\theta(y \mid x)}{\pi_{\mathrm{chosen}}(y \mid x)}.$$

**The variance-based bound.**

Let $\mu = \mathbb{E}_{\pi_\theta}[f]$ and $\sigma^2 = \mathrm{Var}_{\pi_\theta}(f)$. If $\sigma/\mu \ll 1$, then by a second-order Taylor expansion of $\log$ around $\mu$,

$$\Delta = \log \mu - \mathbb{E}\big[\log f\big] \ \leq \ \frac{\sigma^2}{2\,\mu^2}\,.$$

As the fluctuations of $f$ decrease (i.e. $\sigma^2 \to 0$), the Jensen gap $\Delta$ approaches zero. Because $\pi_{\mathrm{chosen}}$ is an energy-based model,

$$f(Y) = \frac{\pi_\theta(y \mid x)}{\pi_{\mathrm{chosen}}(y \mid x)} \ \propto \ \frac{\pi_\theta(y \mid x)}{\pi_{\mathrm{ref}}(y \mid x)} + \log Z(x).$$

That is to say, in order to curb the additional Jensen-gap error we must keep the ratio $\frac{\pi_\theta(y|x)}{\pi_{\mathrm{ref}}(y|x)}$ close to unity—i.e., constrain the policy's deviation from the reference distribution—thereby limiting $\mathrm{Var}(f)$ and tightening the bound; this perspective provides an alternative justification for the ratio-clipping heuristic employed by PPO-style methods.

# F PRELIMINARIES

**Notation** Let $x \in \mathcal{X}$ be a user prompt, $y \in \mathcal{Y}$ a textual response, and $\pi(y \mid x)$ a conditional language model (policy). We denote

- $\pi_{\text{ref}}$ — a fixed reference model (e.g. SFT checkpoint),
- $\pi_\theta$ — the trainable policy with parameters $\theta$,
- $\pi_{\text{chosen}}$ — the human-preferred generator producing positive samples $y_w$, modeled as an energy-based model (see Equation A.1).
- $\pi_{\text{rejection}}$ — the lower-preference generator producing negative samples $y_l$, also instantiated as an energy-based model.
- $\bar{\pi} = \frac{1}{2}(\pi_{\text{chosen}} + \pi_{\text{rejection}})$ — the mixed negative pool used in MINE sampling.

**Reinforcement Learning from Human Feedback.** Given the estimated reward function $r(\mathbf{x}, \mathbf{y})$, dictating the human preferences, RLHF fine-tunes policy $\pi_\theta$ by optimizing the following objective:

$$\max_{\pi_\theta} \mathbb{E}_{\mathbf{y} \sim \pi_\theta(\mathbf{y}|\mathbf{x})} \left[ r(\mathbf{x}, \mathbf{y}) \right] - \beta \mathrm{D}_{\mathrm{KL}} \left[ \pi_\theta(\mathbf{y}|\mathbf{x}) \| \pi_{\text{ref}}(\mathbf{y}|\mathbf{x}) \right], \tag{F.1}$$

where $\beta > 0$ is an appropriate KL penalty coefficient. RLHF typically optimizes the above objective in Equation F.1 using RL algorithms, such as PPO (Schulman et al., 2017). Although RLHF with PPO has achieved remarkable success, the training process of PPO is unstable because of the high variance of the estimates of the policy gradients (Engstrom et al., 2020).

**Reward Modeling.** One standard approach to reward modeling is to fit a reward function $r_\phi(\mathbf{x}, \mathbf{y})$ with the BT preference model in Equation (1). Specifically, the reward function $r_\phi(\mathbf{x}, \mathbf{y})$ can be estimated by maximizing the log-likelihood over preference feedback $(\mathbf{x}, \mathbf{y}_w, \mathbf{y}_l)$:

$$\mathcal{L}_{\mathrm{RM}}(\phi; \mathcal{D}) = \mathbb{E}_{(\mathbf{x}, \mathbf{y}_w, \mathbf{y}_l) \sim \mathcal{D}} \left[ -\log \sigma \left( r_\phi(\mathbf{x}, \mathbf{y}_w) - r_\phi(\mathbf{x}, \mathbf{y}_l) \right) \right]. \tag{F.2}$$

**Mutual Information and the DV Lower Bound** For two random variables $(Y, Z)$ with joint $P_{YZ}$ and marginals $P_Y, P_Z$, the mutual information is $I(Y; Z) = \mathrm{KL}(P_{YZ} \| P_Y P_Z)$. The **Donsker–Varadhan** (DV) variational form writes

$$I(Y; Z) = \sup_{T \in \mathcal{F}} \left( \mathbb{E}_{P_{YZ}}[T] - \log \mathbb{E}_{P_Y P_Z}[e^T] \right), \tag{F.3}$$

where $\mathcal{F}$ is any function class with finite log-moment (Donsker & Varadhan, 1983; Belghazi et al., 2018b). Belghazi et al. (2018a) propose parametrising $T_\phi$ by a neural network and maximising the RHS of equation F.3; the resulting estimator is known as *MINE*.

**Energy-based Models.** Energy-based models (EBMs) (Lecun et al., 2006) define the distribution through an energy function. For $\mathbf{y} \in \mathbb{R}^D$, its probability density can be expressed as follows:

$$p_\theta(\mathbf{y}) = \exp(-E_\theta(\mathbf{y})) / Z_\theta(\mathbf{y}), \tag{F.4}$$

where $E_\theta(\mathbf{y}) : \mathbb{R}^D \to \mathbb{R}$ is the energy function, mapping the data point $\mathbf{y}$ to a scalar, and $Z_\theta(\mathbf{y}) = \sum_{\mathbf{y}} \exp(-E_\theta(\mathbf{y}))$ is the unknown normalization constant (Song & Kingma, 2021).

# G EXPERIMENTAL DETAILS

The source code used in our experiments is available at: `https://anonymous.4open.science/r/MIO-63E6/`

## G.1 TRAINING DETAILS

We follow the general training configurations established in SimPO. During the supervised fine-tuning (SFT) stage, we use a learning rate of $2 \times 10^{-5}$. For both SFT and preference optimization, we set the batch size to 128 and the maximum sequence length to 2048. A cosine learning rate scheduler is

Table 2: The hyperparameter search space for the baselines.

| Method | | Method | |
|---|---|---|---|
| DPO | $\beta \in [0.01, 0.05, 0.1]$ | IPO | $\tau \in [0.01, 0.1, 0.5, 1.0]$ |
| NCA | $\beta \in [0.01, 0.10, 0.15]$ | SLiC | $\lambda \in [0.1, 0.5, 1.0, 10.0]$, |
| | | | $\delta \in [0.1, 0.5, 1.0, 2.0]$ |
| KTO | $\lambda_l = \lambda_w = 1.0$ | SimPO | $\beta \in [2.0, 2.5]$ |
| $\beta \in [0.01, 0.05, 0.1]$ | | $\gamma \in [0.3, 0.5, 1.0, 1.2, 1.4, 1.6]$ | |
| DIL | $\beta \in [0.8, 1.0, 1.2]$ | ORPO | $\beta \in [0.01, 0.05, 0.1]$ |
| CPO | $\beta \in [0.1, 0.01]$ | MIO | $\beta \in [5.0, 7.5, 8.75, 10.0]$ |

applied with a 10% warm-up ratio over a single epoch. We use the Adam optimizer (Kingma & Ba, 2014) in all experiments.

Method-specific hyperparameters are selected according to the search strategy described in SimPO. Each baseline method uses its own dedicated hyperparameter set, as detailed in Table 2. The learning rate for each method is tuned within the range $\{3e{-}7, 5e{-}7, 7e{-}7, 1e{-}6, 5e{-}6\}$.

To mitigate length bias, we normalize the response likelihood by computing the average log-probability of all tokens in the response under the policy model, following SimPO and DIL.

The mistral-7b-sft experiments are conducted using six NVIDIA A6000 GPUs (46GB each), with a total batch size of 384, while the llama and qwen experiments were performed on eight NVIDIA H20 GPUs. We build on the publicly available `alignment-handbook` codebase.

### G.2 THE DETAILS OF DATASETS

**UltraFeedbackBinarized** (Cui et al., 2023a; Tunstall et al., 2023b) is a dataset comprising 64k prompts, each paired with four completions generated by various open-source and proprietary language models. GPT-4 assigns scores to these completions based on criteria such as helpfulness, honesty, and other qualitative metrics. To form binary preference pairs, the completion with the highest average score is selected as the preferred response, while one of the remaining three completions is randomly chosen as the rejected response.

### G.3 EVALUATION TASKS: MATH AND REASONING BENCHMARKS

This section introduces the benchmark used for model evaluation. We follow prior work to evaluate the model fine-tuned on the UltraFeedbackBinarized dataset. Subsequently, we assess its performance on a suite of eight math- and reasoning-oriented tasks, aiming to comprehensively evaluate its quantitative reasoning ability, multi-step problem-solving skills, and domain-specific competence.

**HendrycksMath** (Hendrycks et al., 2021b) consists of 12,500 high school mathematics problems spanning algebra, geometry, and number theory. Each problem is accompanied by a step-by-step solution, targeting rigorous symbolic reasoning.

**MinervaMath** (Lewkowycz et al., 2022b) evaluates quantitative reasoning over STEM domains using university-level mathematics, science, and engineering problems. It emphasizes complex problem solving in structured formats.

**MultiMedQA** (Singhal et al., 2023) is a multi-task benchmark for medical question answering, aggregating datasets from professional exams, clinical research, and consumer health queries. It measures both factual correctness and clinical alignment.

**MathQA** (Amini et al., 2019b) contains over 37k multiple-choice math word problems paired with symbolic execution programs. The dataset encourages interpretable problem solving using operation-based formalisms.

**GSM8K** (Cobbe et al., 2021b) includes 8,500 elementary school math word problems requiring multi-step reasoning. It is widely used to benchmark arithmetic and logical problem-solving capabilities of language models. For this task, we report results using the `exact match` metric under the `flexible-extract` setting, which allows for variations in answer formatting.

**AQuA-RAT** (Cui et al., 2023b) focuses on algebraic word problems from the AQuA-RAT dataset, requiring rationales and supporting steps. It is part of the broader AGIEval benchmark designed to test cognitive ability in foundation models. For this task, we use `normalized accuracy` (`acc_norm`) as the evaluation metric to account for formatting variability in symbolic responses.

**MATH-Hard** is a curated subset from the Hugging Face Open LLM Leaderboard that emphasizes challenging math problems, including multi-hop symbolic reasoning and advanced problem types.

**MuSR** (Sprague et al., 2023) targets multistep soft reasoning problems, typically presented in narrative forms such as logic puzzles and commonsense tasks. It evaluates language models' ability to track state, causality, and inference chains over extended contexts.

# H PROOF OF DV/MINE STARVATION THEOREM

## H.1 SETUP AND ASSUMPTIONS

**Objective.** We adopt the DV/MINE lower bound and the RLHF sampling scheme used in the paper. *DV/MINE objective (up to an additive constant):*

$$I_{\mathrm{DV}}(\theta) = \sup_{\phi} \mathbb{E}_{P_{\pi_\theta \pi_{\mathrm{chosen}}}}[T_\phi(\pi_\theta, \pi_{\mathrm{chosen}})] - \log\left[\mathbb{E}_{P_{\pi_\theta}}\mathbb{E}_{P_{\bar\pi}}[e^{T_\phi(\pi_\theta, \bar\pi)}]\right] \tag{H.1}$$

This is the same modified DV bound and sampling as in Appendix C.1 of the paper.

We will consider two practically important critic families $T_\phi$:

*$\theta$-independent $T$:* Here, $T_\phi$ is independent of $\theta$. We train a separate neural network $\phi$ to estimate the value, which is equivalent to training a reward model in RLHF.

*Log-ratio restricted T function:*

$$T_\theta(\pi_\theta, \pi_{\mathrm{chosen}}) = \log \frac{\pi_\theta(y \mid x)}{\pi_{\mathrm{chosen}}(y \mid x)} + c, \tag{H.2}$$

for some (irrelevant) constant offset $c$. This coincides with the restricted family $\mathcal{F}_{\mathrm{sub}}$ in Appendix C.4 and with the derivations leading to DPO in §2/B.1 of the main text.

**Support assumption.** We also use the paper's support assumption (Appendix A): $\pi_{\mathrm{chosen}}$ and $\pi_{\mathrm{rejection}}$ are energy-based re-weightings of $\pi_{\mathrm{ref}}$ (Eq. (A.1)), hence

$$\pi_{\mathrm{ref}}(y^\star \mid x^\star) = 0 \implies \pi_{\mathrm{chosen}}(y^\star \mid x^\star) = \pi_{\mathrm{rejection}}(y^\star \mid x^\star) = \bar\pi(y^\star \mid x^\star) = 0. \tag{H.3}$$

**Softmax parameterization and the "own-logit" coordinate.** Throughout, we consider the softmax/logit parameterization: for every prompt $x$, there are logits $s_\theta(x, y)$ such that

$$\pi_\theta(y \mid x) = \frac{e^{s_\theta(x,y)}}{\sum_{y'} e^{s_\theta(x,y')}}. \tag{H.4}$$

Let $u$ denote the particular logit coordinate $u := s_\theta(x^\star, y^\star)$. Differentiating the log-softmax gives

$$\frac{\partial}{\partial u} \log \pi_\theta(y \mid x) = \begin{cases} 1 - \pi_\theta(y^\star \mid x^\star), & \text{if } (x, y) = (x^\star, y^\star), \\ -\pi_\theta(y^\star \mid x^\star), & \text{if } x = x^\star, \ y \neq y^\star, \\ 0, & \text{if } x \neq x^\star. \end{cases} \tag{H.5}$$

We analyze the directional derivative along the score direction $\nabla_\theta \log \pi_\theta(y^\star \mid x^\star)$. Under mild regularity (local reparameterization), this is proportional to the partial derivative w.r.t. the "own-logit" $u = s_\theta(x^\star, y^\star)$. Hence it suffices to compute $\partial I_{\mathrm{DV}}(\theta)/\partial u$; the claimed inner product equals this derivative up to a positive scaling.

## H.2 MAIN THEOREM: DV/MINE STARVATION THEOREM

**Theorem H.1** (Gradient starvation for DV/MINE). *Fix $(x^\star, y^\star)$ with $\pi_{\mathrm{ref}}(y^\star \mid x^\star) = 0$ so that* (H.3) *holds. Consider $I_{\mathrm{DV}}(\theta)$ in* (H.1). *Then for any $\theta$ with $\pi_\theta(y^\star \mid x^\star) \geq 0$ (in particular at the boundary $\pi_\theta(y^\star \mid x^\star) = 0$),*

$$\langle \nabla_\theta I_{\mathrm{DV}}(\theta), \nabla_\theta \log \pi_\theta(y^\star \mid x^\star) \rangle = 0,$$

*in each of the following two cases:*

*(a)* $T_\phi$ *does not depend on* $\theta$ *(standard MINE critic), or*

*(b)* $T_\theta(x, y) = \log \frac{\pi_\theta(y|x)}{\pi_{\mathrm{ref}}(y|x)} + c$ *(log-ratio restricted critic from Appendix C.2).*

**Proof.** We differentiate $I_{\mathrm{DV}}(\theta)$ w.r.t. the logit $u = s_\theta(x^\star, y^\star)$ and treat the two cases separately. By the envelope theorem (the inner $\sup_\phi$ can be ignored when differentiating w.r.t. $\theta$ at the maximizer), the derivative equals the $\theta$–partial of the objective at the active critic. This yields

$$\frac{\partial I_{\mathrm{DV}}(\theta)}{\partial u} = \underbrace{\mathbb{E}_{P_{\pi_\theta \pi_{\mathrm{chosen}}}}\left[\frac{\partial T}{\partial u}\right]}_{(A)} - \underbrace{\frac{\mathbb{E}_{P_{\pi_\theta} P_{\bar{\pi}}}\left[e^T \cdot \frac{\partial T}{\partial u}\right]}{\mathbb{E}_{P_{\pi_\theta} P_{\bar{\pi}}}\left[e^T\right]}}_{(B)}, \tag{H.6}$$

*Case (a): $\theta$-independent critic.* Then $\partial T/\partial u \equiv 0$, hence both $(A)$ and $(B)$ vanish, and $\partial I_{\mathrm{DV}}/\partial u = 0$. Therefore the claimed inner product is $0$.

*Case (b): T is a log-ratio critic.* Here

$$\frac{\partial T}{\partial u} = \frac{\partial T}{\partial u}(\pi_\theta, \pi_{\mathrm{chosen}}) = \frac{\partial}{\partial u} \log \pi_\theta(y \mid x) = \mathbf{1}\{x = x^\star\}\left(\mathbf{1}\{y = y^\star\} - \pi_\theta(y^\star \mid x^\star)\right) \quad \text{by (H.5).} \tag{H.7}$$

Using *(H.7)* in $(A)$ and the fact $\pi_{\mathrm{chosen}}(y^\star \mid x^\star) = 0$ *(H.3)*, we get

$$(A) = \mathbb{E}_{x \sim D}\, \mathbf{1}\{x = x^\star\}\, \mathbb{E}_{y^+ \sim \pi_{\mathrm{chosen}}(\cdot|x^\star)}\left[\mathbf{1}\{y^+ = y^\star\} - \pi_\theta(y^\star \mid x^\star)\right] = -\pi_\theta(y^\star \mid x^\star). \tag{H.8}$$

Similarly, using *(H.7)* in $(B)$ and $\bar{\pi}(y^\star \mid x^\star) = 0$ *(H.3)*,

$$(B) = \frac{\mathbb{E}_{x \sim D}\, \mathbf{1}\{x = x^\star\}\, \mathbb{E}_{y \sim \bar{\pi}(\cdot|x^\star)}\left[e^{T(x^\star, y)}\left(\mathbf{1}\{y = y^\star\} - \pi_\theta(y^\star \mid x^\star)\right)\right]}{\mathbb{E}_{x \sim D}\, \mathbf{1}\{x = x^\star\}\, \mathbb{E}_{y \sim \bar{\pi}(\cdot|x^\star)}\left[e^{T(x^\star, y)}\right]} = -\pi_\theta(y^\star \mid x^\star). \tag{H.9}$$

Plugging *(H.8)–(H.9)* into *(H.6)* gives $\frac{\partial I_{\mathrm{DV}}}{\partial u} = (-\pi_\theta) - (-\pi_\theta) = 0$. Again, the claimed inner product is zero. $\square$

### H.3 Quantitative decay near the boundary

Theorem H.1 shows exact cancellation for the two common critic forms. One may also ask for quantitative bounds when the critic depends on $\theta$ more generally.

**Theorem H.2** (Linear decay near zero under Lipschitz critics)**.** *Suppose the active critic can be written as $T_\theta(\pi_\theta, \pi_{chosen}) = G(x, y, \log \pi_\theta(y \mid x))$ where $G$ is Lipschitz in its third argument with constant $L$ (uniformly in $(x, y)$). Under the support condition* (H.3)*, for any $\theta$,*

$$\left|\langle \nabla_\theta I_{\mathrm{DV}}(\theta), \nabla_\theta \log \pi_\theta(y^\star \mid x^\star)\rangle\right| \leq 2L\, \pi_\theta(y^\star \mid x^\star). \tag{H.10}$$

*In particular, as $\pi_\theta(y^\star \mid x^\star) \downarrow 0$, the directional derivative vanishes at least linearly.*

**Proof.** As in *(H.6)*, $\partial I_{\mathrm{DV}}/\partial u = \mathbb{E}_{P_{\pi_\theta \pi_{\mathrm{chosen}}}}[\partial_u T_\theta] - \frac{\mathbb{E}_{P_{\pi_\theta} P_{\bar{\pi}}}[e^{T_\theta} \partial_u T_\theta]}{\mathbb{E}_{P_{\pi_\theta} P_{\bar{\pi}}}[e^{T_\theta}]}$. By the chain rule, $\partial_u T_\theta = (\partial_3 G) \cdot \partial_u \log \pi_\theta(y \mid x)$. The Lipschitz assumption yields $|\partial_3 G| \leq L$ almost everywhere; combine this with *(H.5)* and *(H.3)* to get

$$\left|\mathbb{E}_{P_{\pi_\theta \pi_{\mathrm{chosen}}}}[\partial_u T_\theta]\right| \leq L\, \mathbb{E}_x\, \mathbf{1}\{x = x^\star\}\, \mathbb{E}_{y^+ \sim \pi_{\mathrm{chosen}}(\cdot|x^\star)}\left[\left|\mathbf{1}\{y^+ = y^\star\} - \pi_\theta(y^\star \mid x^\star)\right|\right] = L\, \pi_\theta(y^\star \mid x^\star),$$

and similarly

$$\left|\frac{\mathbb{E}_{P_{\pi_\theta} P_{\bar{\pi}}}[e^{T_\theta} \partial_u T_\theta]}{\mathbb{E}_{P_{\pi_\theta} P_{\bar{\pi}}}[e^{T_\theta}]}\right| \leq L\, \pi_\theta(y^\star \mid x^\star).$$

Triangle inequality then implies *(H.10)*. $\square$

**Consequences.** For the log-ratio critic (Theorem H.1(b)), we have $L = 1$ and the two terms are exactly equal, yielding exact zero (stronger than *(H.10)*). For any $\theta$-independent critic (Theorem H.1(a)), $\partial_3 G \equiv 0$, hence the bound is trivially zero. For other smooth $\theta$-dependent critics, *(H.10)* shows a linear decay of the directional derivative as $\pi_\theta(y^\star \mid x^\star) \to 0$, formalizing the "the closer to the boundary, the weaker the signal" intuition.

# I MIO LEVERAGES THE REWARD SIGNAL MORE EFFECTIVELY THAN NCA-PAIR.

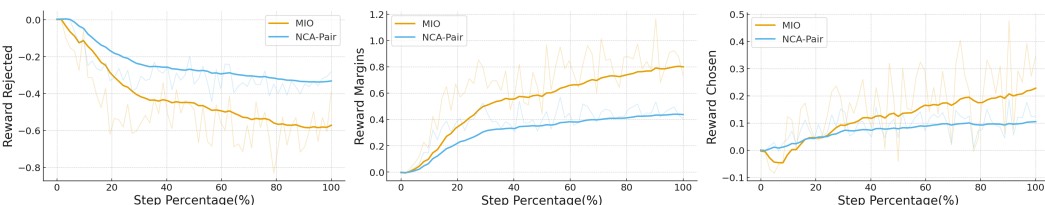

Figure 5: MIO leverages the reward signal more effectively than NCA-Pair.

Figure 5 presents a direct comparison between MIO and NCA-Pair in terms of reward margin and related metrics. The results show that MIO achieves a greater increase in both the chosen reward and the reward margin, as well as a larger decrease in the rejected reward, compared to NCA-Pair.

# J LLM USAGE

Large Language Models (LLMs) were used solely to assist with writing and editing of this manuscript—for tasks such as wording refinement, grammar checking, and improving readability. LLMs were not involved in research ideation, methodology, data collection, analysis, or result selection. All scientific content and claims were produced and verified by the authors, who take full responsibility for the manuscript. We ensured that any LLM-assisted text complies with ethical guidelines and does not constitute plagiarism or scientific misconduct.

