# OpenReview forum: "The Hidden Link Between RLHF and Contrastive Learning"
_ICLR.cc/2026/Conference — Submitted to ICLR 2026_

### Official Review · Reviewer_YnUp · 2025-10-27

**Soundness:** 2
**Presentation:** 1
**Contribution:** 2
**Rating:** 2
**Confidence:** 4

**Summary:**

This paper proposes to interpret Reinforcement Learning from Human Feedback (RLHF) and Direct Preference Optimization (DPO) from a mutual-information (MI) perspective. The authors claim both can be viewed as instances of contrastive learning under the Donsker–Varadhan (DV) lower bound. The authors further propose a new Jensen–Shannon–based estimator (termed Mutual Information Optimization, MIO) to alleviate the gradient starvation issues observed in PPO/DPO. Experiments on reasoning and math benchmarks suggest that MIO improves stability and performance.

**Strengths:**

The paper attempts to unify RLHF, DPO, and contrastive learning within a single theoretical framework, which is conceptually interesting.

**Weaknesses:**

1. Clarity and Writing Quality. The manuscript is extremely difficult to read.
Key notations such as $y_l$, $\beta$ $I_DV$ $T$ are not explained or not detailed introduced when they first appear.
Many formulas are introduced abruptly, without intermediate derivations For example, it is not clear how (7) is obtained by (5) and (6).
Language contains numerous grammatical errors and inconsistent style. For example, $T$ and T are used alternatively. Line 146 has two predicates.

2. Lack of Theoretical Rigor. Theorems are not presented with assumptions as presented in the appendix. It would be more regions if the author properly adjust the structure of the theoretical notations, assumptions and presentations.

**Questions:**

Why instead of retraining a separate neural network $T_\phi$, the authors constrain the T to the log-ratio sub-family?

---

### Official Review · Reviewer_tWrk · 2025-10-31

**Soundness:** 2
**Presentation:** 3
**Contribution:** 2
**Rating:** 4
**Confidence:** 3

**Summary:**

This paper presents a unified theoretical framework that re-interprets RLHF and DPO as forms of contrastive learning through the lens of mutual information (MI) maximization. The authors identify the Donsker-Varadhan (DV) MI estimator as a root cause for key failure modes in these methods and propose a new algorithm, Mutual Information Optimization (MIO), which replaces the DV estimator with a more stable Jensen-Shannon (JS) divergence-based estimator. MIO is demonstrated to reduce training instability and performance degradation, achieving competitive results on several reasoning benchmarks.

**Strengths:**

1. The paper proposes a unifying theoretical framework: a mutual information-based perspective that unifies RLHF and DPO with contrastive learning.

2. The empirical validation uses a toy model to isolate the failure mode, large-scale fine-tuning on multiple base models, and compares against some representative baselines across several reasoning benchmarks.

3. The paper is well-structured and the authors have provided detailed experimental setups.

**Weaknesses:**

1. The first concern is that this work only replaces an existing information form (DV) with another existing information form (JS). Moreover, to simplify deduction and calculation, it restricts the critic to the log ratio family and yields only an even looser *surrogate* bound (equiv. to Eq. 15). Then the learning scheme is to optimize this looser surrogate bound, which is a plain half-mix of the "chosen" and "rejection" informations. It is difficult to extract a significant contribution from Eq. 15.

2. While the DPO loss resembles a contrastive objective that seeks to increase the log-likelihood margin between the chosen ($\pi(y^c|x)$) and rejected ($\pi(y^r|x)$) responses, the technical equivalence is not straightforward and relies on the explicit reward function derived from the Bradley-Terry model. The DPO loss is derived by substituting this reward function into the RL objective and then finding the optimal policy without explicitly maximizing MI. It is unclear whether this treatment is reliable to replace the explicit maximization of MI. Besides, true MI maximization in a sequential setting (like an LLM generating tokens) would involve a temporal mutual information or a specific causal structure (e.g., maximizing $I(\text{Prompt}, \text{Chosen Response})$ while minimizing $I(\text{Prompt}, \text{Rejected Response})$). The DPO loss is a static, end-to-end objective applied over the full response sequence. Asserting that this static objective maximizes MI over the sequential generation process is an overclaim, especially since it does not explicitly manage the token-level dependencies $I(y_t; y_{<t})$ in the way a true generative model MI bound will.

3. Reward over-optimization (or "reward hacking") occurs because the learned Reward Model (RM), $R(x, y)$, is an imperfect proxy for true human preferences. The policy, $\pi_\theta$, learns to exploit the *defects* of $R(x, y)$ by finding high-scoring, unnatural generations. The MIO objective replaces the DV lower bound with the JS MI estimator. This change affects the training stability and fidelity of the implicit contrastive loss. It does not inherently guarantee that the resulting policy will avoid exploiting spurious correlations within the preference data that caused the RM's misgeneralization in the first place. Since DPO and its variants (like MIO, which is fundamentally a DPO-style contrastive loss) still rely on the relative preference ranking $(\pi(y^c|x) / \pi(y^r|x))$, they are still susceptible to issues where the preferred and rejected completions only differ slightly (low edit distance). In such cases, the training only forces a small change in the log-probability margin, and the model might still develop misaligned behaviors that are not penalized by the contrastive loss but would be penalized by a truly perfect RM.

4. While the paper compares to DPO, it neglects comparisons against Proximal Policy Optimization (PPO)—the classical RLHF method—or modern PPO variants. Since RLHF is also cast as an MI maximization, a robust comparison must show that MIO is superior to PPO (or at least competitive while being simpler) across various tasks, not just DPO. Additionally, comparisons with other DPO variants like IPO (Identity Preference Optimization, [1]) and KTO (Kahneman-Tversky Optimization, [2]) are also missing. These missing experiments should be conducted to further examine the effectiveness of the proposed MIO.


[1] Azar, M. G., Guo, Z. D., Piot, B., Munos, R., Rowland, M., Valko, M., \& Calandriello, D. (2024, April). A general theoretical paradigm to understand learning from human preferences. In International Conference on Artificial Intelligence and Statistics (pp. 4447-4455). PMLR.

[2] Kawin Ethayarajh, Winnie Xu, Niklas Muennighoff, Dan Jurafsky, and Douwe Kiela. 2024. Model alignment as prospect theoretic optimization. In Proceedings of the 41st International Conference on Machine Learning (ICML'24), Vol. 235. JMLR.org, Article 504, 12634–12651.

The current form of the paper is not enough convincing in motivation, methodology, and experiment.

**Questions:**

See weaknesses.

---

### Official Review · Reviewer_kC47 · 2025-10-31

**Soundness:** 3
**Presentation:** 2
**Contribution:** 1
**Rating:** 4
**Confidence:** 2

**Summary:**

This work proposes a novel perspective that frames RLHF and DPO as a unified form of mutual information based contrastive learning over the distributions of chosen and rejected responses. Based on this view, this work explains why reasoning paths that are not sampled by the base model remain inaccessible even after RL training, and why DPO updates the small policy values too slowly. Then this work proposes Mutual Information Optimization (MIO) approach to solve these issues, by replacing the mutual information lower bound with Jensen–Shannon (JS) estimator of the mutual information.

**Strengths:**

The claims in the summary above look strong and novel, as it theoretically explains the failure of the existing RLHF and DPO approaches, and proposes a new method to solve this also with theoretical foundation. The experimental details look comprehensive to reproduce the results. The experiments cover many existing models and benchmarks. The improvements in the experimental results by the proposed MIO method look significant in some cases.

**Weaknesses:**

The math formulations are very unclear. Specifically, the vagueness of notations especially $T_{\phi}$ persists throughout the whole paper. I list the main unclear points in the questions below.

**Questions:**

(1) In Eq. (3), $T_{\phi}$ looks vague. Is $T_{\phi}$ a function from the neural network parameter $\phi$ to a scalar? Where does the randomness of $T_{\phi}$ come from? What is the role of $T_{\phi}$?

(2) Why is optimizing Eq. (5) better than optimizing Eq. (4)?

(3) $M$, $N$ and $T _ {\phi}(\pi_{\theta},\pi)$ for $\pi\in\{\pi _ {\rm chosen}, \pi _ {\rm rejection}\}$ needs to be defined right after Eq. (6). Also, should $T _ {\phi}(\pi_{\theta},\pi)$ be something like $T _ {\phi, i}(\pi_{\theta},\pi)$? Seems there are two datasets with sizes $M$ and $N$ respectively? If yes, what samples do the two datasets have?

(4) From Eq. (9) $T$ actually depends on $\theta$, so why not using $T _ {\theta}$ instead of $T _ {\phi}$?

(5) What does “sp” mean in Eq. (12)?

(6) When interpreting Propositions 3.1 and 3.2, note that $\pi^+$ and $\pi^-$ are constrained in the policy space $\Pi=${$\pi(y|x)\ge 0$ and $\sum_y \pi(y|x)=1$}. Hence, we cannot directly apply gradient descent, but may need projection like $\pi _ {t+1}^+={\rm proj} _ {\Pi}\Big[\pi_t^+ - \alpha\frac{\partial\ell _ {\rm MIO}}{\partial \pi_t^+}\Big]$.

(7) Table 1 caption could state the metrics used in each benchmark.

(8) In Figure 3, why do not we want the reward rejected to decrease as much as possible?

(9) Typos—
End of page 1: tuining $\to$ tuning.
Line 90: “We provide”
In line 279, it seems that $\alpha=z=1$?

---

### Official Review · Reviewer_fbYr · 2025-11-01

**Soundness:** 3
**Presentation:** 3
**Contribution:** 3
**Rating:** 6
**Confidence:** 2

**Summary:**

This paper recasts alignment methods like RLHF and DPO as forms of contrastive learning that implicitly optimize the Donsker-Varadhan (DV) mutual information (MI) estimator. The authors provide a theoretical explanation, the "DV/MINE Starvation Theorem," arguing this specific estimator is the root cause for observed failure modes, such as DPO's "synchronous collapse" and the inability of RLHF to discover new reasoning paths when a response probability is near zero. To solve this, the paper introduces Mutual Information Optimization (MIO), a new alignment objective that replaces the unstable DV bound with the more robust Jensen-Shannon (JS) MI estimator. Theoretical analysis and empirical results on eight math and reasoning benchmarks demonstrate that MIO avoids this collapse, offering greater training stability and achieving superior performance.

**Strengths:**

- The work originally connects popular alignment algorithms (RLHF, DPO) to contrastive learning via mutual information (MI) maximization. The key insight—attributing their observed failure modes not just to the objective but to the specific MI estimator they implicitly use (the Donsker-Varadhan bound) —is highly creative.
- The paper is of high quality, providing rigorous theoretical support for its claims. It introduces the "DV/MINE Starvation Theorem"  to formalize why DV-based methods fail when response probabilities are near zero.
- The paper is well-written. It organizes its contributions around three "Major Questions" and provides direct, explicit answers, guiding the reader through its complex argument.

**Weaknesses:**

The authors demonstrate MIO's strong performance in Table 1, but almost exclusively on a suite of mathematical and reasoning benchmarks. This is the exact domain the paper identifies as a key failure point for DPO. While this supports the specific claim about fixing DPO's performance degradation on these tasks, it fails to substantiate MIO as a superior general-purpose alignment method. The evaluation is missing key standard benchmarks needed to assess the full picture and potential trade-offs, such as:

- General instruction-following (e.g., AlpacaEval, MT-Bench)
- Broad multi-domain knowledge (e.g., MMLU)
- Other alignment criteria like truthfulness (e.g., TruthfulQA)

**Questions:**

Given the evaluation's focus on math and reasoning tasks, have the authors conducted any experiments on broader, general-purpose benchmarks such as MMLU, AlpacaEval, or TruthfulQA? If so, could these results be shared to compare MIO against DPO and SFT baseline on these other axes?

---

### Meta-Review · Area_Chair_7HwZ · 2025-12-21

**Summary:**

The common issues lied in unclear and confusing math derivation, missing key standard benchmarks and comparisons with other DPO variants.

The authors did not reply to the review comments.

It has to be rejected.

**Reviewer Concerns:**

No rebuttals.

**Reviewer Scores:**

It received 4 reviews with the average score 4.

No discussions for no rebuttals.

---

### Decision · Program_Chairs · 2026-01-26

Reject